# Target Recognition in SAR Images by Deep Learning with Training Data Augmentation

**DOI:** 10.3390/s23020941

**Published:** 2023-01-13

**Authors:** Zhe Geng, Ying Xu, Bei-Ning Wang, Xiang Yu, Dai-Yin Zhu, Gong Zhang

**Affiliations:** 1Key Laboratory of Radar Imaging and Microwave Photonics, Nanjing University of Aeronautics and Astronautics, Nanjing 211106, China; 2School of Computer Engineering, Nanjing Institute of Technology, Nanjing 211167, China

**Keywords:** automatic target recognition, deep learning, sparse representation, synthetic aperture radar, training data augmentation

## Abstract

Mass production of high-quality synthetic SAR training imagery is essential for boosting the performance of deep-learning (DL)-based SAR automatic target recognition (ATR) algorithms in an open-world environment. To address this problem, we exploit both the widely used Moving and Stationary Target Acquisition and Recognition (MSTAR) SAR dataset and the Synthetic and Measured Paired Labeled Experiment (SAMPLE) dataset, which consists of selected samples from the MSTAR dataset and their computer-generated synthetic counterparts. A series of data augmentation experiments are carried out. First, the sparsity of the scattering centers of the targets is exploited for new target pose synthesis. Additionally, training data with various clutter backgrounds are synthesized via clutter transfer, so that the neural networks are better prepared to cope with background changes in the test samples. To effectively augment the synthetic SAR imagery in the SAMPLE dataset, a novel contrast-based data augmentation technique is proposed. To improve the robustness of neural networks against out-of-distribution (OOD) samples, the SAR images of ground military vehicles collected by the self-developed MiniSAR system are used as the training data for the adversarial outlier exposure procedure. Simulation results show that the proposed data augmentation methods are effective in improving both the target classification accuracy and the OOD detection performance. The purpose of this work is to establish the foundation for large-scale, open-field implementation of DL-based SAR-ATR systems, which is not only of great value in the sense of theoretical research, but is also potentially meaningful in the aspect of military application.

## 1. Introduction

In recent years, with interest in artificial intelligence (AI) soaring, synthetic aperture radar (SAR) automatic target recognition (ATR) with deep neural networks (DNNs) has attracted the attention of researchers in academia from all over the world [1,2]. However, the SAR images obtained in field experiments that have been manually labeled and can be used for the DNN-training are very limited. One of the most commonly used measured SAR imagery datasets is the Moving and Stationary Target Acquisition and Recognition (MSTAR) SAR dataset, which was collected by the Defense Advanced Research Projects Agency (DARPA) and the Air Force Research Laboratory (AFRL) between 1995 and 1997 [3]. The publicly released version consists of 20,000 SAR image chips covering the self-propelled howitzer (2S1), the armored personnel carriers (BMP2, BRDM2, BTR60, and BTR70), the anti-aircraft gun (ZSU23-4), the main battle tanks (T62 and T72), the truck (ZIL131), and the bulldozer (D7). Another important SAR imagery dataset covering the same set of military vehicle targets is the QinetiQ (UK) SAR data recorded with the Enhanced Surveillance Radar (ESR) in 2001, which was used to double-check the performance of the SAR-ATR algorithms derived based on the MSTAR dataset. Compared with the MSTAR dataset, the SAR image samples in the QinetiQ dataset are more realistic and feature neither centered targets nor sharp-edged shadow regions [4,5]. With the QinetiQ dataset, Schumacher et al. proved that both the clutter and the shadow region play important roles in SAR target classification, and that carrying out performance evaluation for the SAR-ATR algorithms based solely on the limited samples from the MSTAR dataset is potentially biased.

In 2021, Kechagias-Stamatis et al. pointed out that since the backgrounds for the training and testing images in the MSTAR database are highly correlated and all the true labels for the test images are known a priori to researchers, a new benchmarking dataset containing an appropriate number of images with publicly unknown class labels is needed to evaluate the true effectiveness of the ATR systems proposed in research papers [6]. To illustrate the effects of clutter background, some representative SAR images in the MSTAR dataset downloaded from the official website (https://www.sdms.afrl.af.mil, accessed on 28 November 2022) are shown in Figure 1, which depict 2S1, BRDM2, and ZSU23 observed at elevation angles of El = 17° and El = 30°. The website offers two types of files: .JEPG images (for DISPLAY purposes) and the raw data files (contains full information and can be converted to any type of images of any quality). The .JEPG images are widely used by researchers in countries having difficulties obtaining access to the official website, and are illustrated in the two image blocks on the left of Figure 1. It can be seen that for El = 30°, the backgrounds of BRDM2 are much brighter than those of 2S1. With specially tuned and enhanced ResNet18, an average classification accuracy of **97.2%** was achieved in 10 iterations. This result might seem promising. However, for neural networks, the parameters are automatically adjusted to achieve higher accuracy. As a result, the high accuracy was much likely contributed by the false feature of “brightness of clutter backgrounds”. To test this theory, we carried out another experiment: the same images were used for training, while the images manually generated from the raw data files with contrast-balancing are adopted as the new test images. As expected, the accuracy of the same network immediately dropped to **65.3%**. It is as if the neural network learned from the training data that “the brighter ones are more likely to be BRDM2”, but cannot see this pattern in the well-balanced test data. However, if we use images with contrast-balancing as our training data from the very beginning, the neural network learns to focus on the “real features” and we achieve an average 10-iteration accuracy of **88.5%** (with the highest/lowest accuracy being 93.2%/85.4%). 

Actually, to realize wide application of DNN-based SAR-ATR algorithms in the open-world environment, where a rich variety of vehicle targets could be present, high-quality synthetic SAR imagery generated with computational electromagnetic (CEM) methods based on the computer-aided design (CAD) models of the potential targets would be of great help. In [7,8], Lewis et al. presented the synthetic and measured paired–labeled experiment (SAMPLE) dataset, which is complementary to the MSTAR dataset and covers the same types of targets. According to [7], the SAMPLE dataset has been in use for many years, and has led to several important research works on transfer learning, neural network modeling, and generative adversarial networks (GAN) by researchers from the Massachusetts Institute of Technology, Georgia Institute of Technology, and AFRL [9,10]. The great value of the SAMPLE dataset for the next phase of development in the research field of SAR-ATR has also been emphasized in the review article by Blasch et al. [11]. However, the publicly released version of the SAMPLE dataset contains only 806 training images and 539 testing images, which is far from enough for the development of DNNs. Moreover, the backgrounds of the synthetic imagery in the SAMPLE dataset are obviously different from the measured ones, which exhibit much higher noise and clutter levels (see Table 1). Another important research paper on CEM-aided synthetic SAR imagery generation worth noting is [12], where Yoo and Kim exploited indirect information (i.e., 2D drawings, multi-aspect photos, and video clips) regarding the targets of interest in the case when the detailed 3D blueprints of the targets are unavailable. Unfortunately, Yoo and Kim elect not to share the dataset they generated with the public. Additionally, just like the SAMPLE dataset, there is an apparent difference between the measured and the synthetic SAR imagery (see Figures 13 and 14 in [12]).

In [13,14,15], several classic optical image augmentation methods are leveraged to compensate for the limited labeled SAR imagery for DNN training, which include noise-adding and image rotation, while in [8,16] the GAN is employed to fulfill the same purpose. Although the data augmentation methods mentioned above have led to improved classification accuracy evaluated by the MSTAR/SAMPLE dataset [17], these methods are not effective in modeling the varying SAR image features caused by the actual changes in radar parameters, observation angle variations, target articulations, and the signal to clutter-plus-noise ratio (SCNR). Unlike the optical imagery, a SAR image depicts the electromagnetic scattering characteristics of the target of interest and its surroundings based on the measurements made by the radar system. Although the resolution of SAR images does not deteriorate with distance, the information content per pixel is very limited and depends highly on the radar waveform properties, the observation angle, the SCNR, and the SAR imaging algorithms [18]. As a result, addressing the technical challenges in SAR-ATR by borrowing completely from the field of optical image recognition is actually trying to oversimplify a complicated problem. For example, when training a DNN to recognize optical images containing “handbag”, it is perfectly valid to use GAN to generate training images featuring handbags with different color and styles based on the rich and detailed information provided by a dozen of images featuring handbags. However, it is not appropriate to rely solely on the GAN to synthesize large numbers of SAR images featuring military/civilian vehicles in the same way without even modeling the target echo signal based on the radar parameters and the observation scene. Recognizing the problems mentioned above, physics-based, high-quality SAR data augmentation has become a new trend. For example, Agarwal et al. proposed a novel method to exploit the sparsity of the scattering centers for data interpolation in the phase-history domain, which exhibited promising preliminary results [19].

The majority of the DNN-based SAR-ATR algorithms exploit the convolutional neural network (CNN) structure, which has made impressive achievement in the field of optical imagery classification. However, unlike the neural networks trained for RGB image recognition, which can contain as many as 1000 layers [20], the CNN algorithms for SAR-ATR are relatively “shallow” since the information contained in each pixel is very limited [21]. In [15], Inkawhich et al. compared the performance of small, medium, and large DNN architectures, which are exemplified by SMPL (a small LeNet-style network [7]), ResNet18, and Wide-ResNet18, respectively, and proved that large architecture does not lead to superior SAR-ATR accuracy. Therefore, in this work, we concentrate on developing data augmentation techniques for light-weighted CNNs. Specifically, we use only 20–30 measured data samples for each type of target included in the MSTAR dataset and achieve a high accuracy of 97.6% with the SMPL network by employing the data augmentation technique based on phase interpolation. To evaluate the effect of clutter background on SAR target classification, we leverage the clutter data in the MSTAR dataset, which are segmented into 1159 clutter chips of size 128 × 128 and divided into two distinctive pools for training and testing. New training and test image samples are generated by implementing the clutter transfer technique, based on which the robustness of the target classification algorithms in clutters with different levels of homogeneity is evaluated. Simulation results show that the target classification performance of the existing algorithms degrades noticeably if we employ the original MSTAR image samples for training and apply clutter transfer to the test samples only. However, by introducing training data with various clutter backgrounds, the algorithm is robust against the background changes in the test samples.

To effectively augment the synthetic SAR image samples in the SAMPLE dataset, we propose a novel contrast-based training data augmentation technique. Specifically, grayscale images with multiple different contrast levels are generated with the complex image data in “.mat” format. Simulation results show that, by employing 100% synthetic SAR images with properly set contrast levels for network training, a 10-class classification accuracy of 95% can be achieved by ResNet18. The main reason behind the success is that the contrast levels between the target and the background of the measured and the synthetic SAR images from the SAMPLE dataset are very different. By varying the contrast levels, we are actually teaching the network that “the background is irrelevant”. Unfortunately, it also means that if the SAR images involved in network training are 100% real measured data, the contrast-based data augmentation method will not be effective.

To improve the robustness of the SAR-ATR algorithms in open-world environments, various types of out-of-distribution (OOD) samples are considered. The same problem was previously studied in [17], where 59,535 image samples from the SAR-ship dataset proposed in [22] are used as the outlier exposure (OE) training data for the adversarial outlier exposure (advOE) procedure. In this work, we introduce another SAR image dataset for advOE training, the Mini-SAR dataset, which consists of SAR images collected by the X-band Mini-SAR developed by Nanjing University of Aeronautics and Astronautics (NUAA-MiniSAR) [23] for ground military vehicle targets. In numerical simulations, 2491 SAR images from the SAR-ship dataset and the MiniSAR dataset are used as the OE training data, which is only 4% of the training samples used in [17]. To evaluate the effect of clutter background on the OOD detection performance, two OOD test datasets are employed: the MSTAR-O and the MSTAR-P. Both datasets consist of 1290 SAR images of five types of targets (BRDM2, BTR60, D7, T62, ZIL131) from the MSTAR dataset. However, the image samples in MSTAR-O are generated based on the official MSTAR raw data, while MSTAR-P consists of image samples downloaded from a website in the public domain, which feature a blurred shadow region and partially eliminated clutter background.

The major contributions are summarized as follows:The effectiveness of the phase-interpolation-based training data augmentation technique is demonstrated with the MSTAR dataset, and a novel contrast-based method is proposed to augment the synthetic training samples in the SAMPLE dataset.The effect of the clutter background on SAR target classification is evaluated by exploiting the clutter data in the MSTAR dataset. It is shown that by introducing training data with various clutter backgrounds, the algorithm is robust against the background changes in the test samples.By using the MiniSAR images as the OE training samples, diverse CNN models are designed and trained to detect OOD samples from the MSTAR-O and the MSTAR-P datasets. It is shown that the contrast-based data augmentation method is also effective in improving the OOD sample detection performance.

The structure of this work is illustrated in Figure 2. The rest of this work is organized as follows. In Section 2, the process of data augmentation via interpolation in the phase-history domain and clutter transfer is presented. In Section 3, effective training methods are proposed for DNN to achieve an outstanding in-distribution (ID) sample classification performance while accurately rejecting the OOD samples. In Section 4, experimental results are provided to demonstrate the performance of the proposed data augmentation methods and the effectiveness of the OOD detection algorithms. Some final remarks are offered in Section 5.

## 2. Data Augmentation via Interpolation in the Phase-History Domain and Clutter Transfer

Leveraging hundreds of open-source training datasets containing millions of optical images featuring objects of interest in numerous places under various light conditions (e.g., ImageNet, LSUN, etc.), the effect of an object’s surroundings on target recognition/classification has been thoroughly investigated, and the DNNs for image recognition have been trained to cope with varying backgrounds and potential camouflage. In contrast, since the measured and labeled SAR images that could be used for DNN training are very limited, the robustness of the SAR-ATR algorithms in the open-world environment still needs improvement. A typical SAR image of a ground-vehicle target consists of two parts: the target and the clutter. Note that although the shadow region is prominent in the SAR images from the MSTAR dataset, it is not a common feature shared by all the SAR images depicting ground targets (e.g., images from the QinetiQ dataset and the MiniSAR dataset). However, most existing research on data augmentation are focused on the *target*, while leaving the *clutter* unattended or even “completely removed” via target-masking technique. To enhance the robustness of the SAR-ATR algorithms against heavy clutter of high heterogeneity, we consider both target and clutter in data augmentation and network training. Accordingly, this section is divided into two subsections. In Section 2.1, the phase-interpolation-based SAR-target-feature synthesis technique is introduced. In Section 2.2, a novel clutter transfer technique is presented by exploiting the clutter data collected during the MSTAR mission.

### 2.1. Data Augmentation via Interpolation

For physics-based SAR image dataset augmentation, we adopt the method proposed in Agarwal et al. [19] and exploit the scattering centers’ spatial sparsity. The working flowchart of interpolation in the phase-history domain is presented in Figure 3**.** To begin, a sparse-representation-based phase-history (PH) model is estimated based on the PH data extracted from a given SAR image. Suppose that a complex target consists of *K_c_* dominant scattering centers and the corresponding spatial coordinates and scattering coefficients are {(*x_k_*, *y_k_*)} and {*h_k_* (*θ*, *ϕ*)}, *k* = 1, …, *K_c_*, respectively, with *θ* and *ϕ* representing azimuth and the elevation angle, respectively. The (*i*, *m*)-th element of the phase-history matrix **S**(**θ***,***f**) ∈ℂNθ×M is given by
(1)s(i,m)=n(i,m)+∑k=1Kchk(θi,ϕ)×exp(−j4πfmcos(ϕ)c(xkcos(θi)+yksin(θi))),
where *n*(*i, m*) is the measurement noise; *θ_i_*, *i* = 1, …, *N_θ_* are the azimuth angles; and *f_m_*, *m* = 1, …, *M* are the illuminating frequencies such that *M* = 2*BL*/*c*, with *B*, *L,* and *c* denoting the bandwidth of the transmitted pulse, the side length of the square-shaped area of interest centered around the target, and the speed of light, respectively. It is assumed that {*h_k_* (*θ_i_*, *ϕ*)}, *i =* 1, …, *N_θ_* can be represented with a basis set Ψ∈ℂNθ×D as
(2)hk(θi,ϕ)=∑ν=1Dcν,kψν(θi)+ϵP
where ψν(θi) is the (*ν, i*)*-th* element of **Ψ** and ϵP is the estimation error. After PH domain interpolation, the PH measurements are converted back to an SAR image using the overlapping subapertures spanning 3° in the azimuth domain, with Taylor window applied for sidelobe control.

Some preliminary results obtained with this method are shown in Figure 4. In this example, four subpixel shifts in four directions are performed on each real SAR image sample (i.e., 1/2 pixel displacement in range/cross-range direction), and subsequently each shifted image is used to generate 48 extrapolated samples. The synthetic samples at an azimuth angle θ° (i.e., Az = θ°) are obtained from the real ones corresponding to Az = θ° + 5°. For example, the image titled “Syn. Az = 301°” in Figure 4a is one of the 49 × 4 = 196 synthetic SAR images generated according to the real SAR image titled as “Real Az = 306°”. It can be seen that when there are only a few dominant scattering points, the simulated SAR images are very similar to the real ones. However, during the experiments, we also noticed that when the target consists of a number of strong scattering points, the difference between the simulated and the real SAR images becomes more striking, which might be due to either the scintillation effects or the robustness of the limited persistence sparse modeling approach when handling complex targets. In Section 4, we demonstrate that by exploiting the phase-interpolation-based data augmentation technique, only 20–30 measured data samples for each type of target are needed by SMPL to achieve a MSTAR 10-class classification accuracy of 97.6%.

### 2.2. Data Augmentation via Clutter Transfer

The clutter data collected during the MSTAR mission can be segmented into 1159 clutter chips of size 128 × 128. Each clutter chip belongs to one of the six categories: “C1” for “cultural isolated object” (e.g., small building, 345 samples); “C2” for “natural isolated object” (e.g., tree, 310 samples); “C3” for “cultural edge/corner” (e.g., roads, 189 samples); “C4” for “natural edge/corner” (e.g., streams, 73 samples); “C5” for “cultural homogeneous area” (e.g., parking lot, 122 samples); “C6” for “natural homogeneous area” (e.g., grass field, 120 samples). To begin, we select 175 most representative clutter chips and divide them into two distinctive pools: a “training pool” and a “test pool”. Afterward, the SARBake pixelwise annotation results for the SAR images in the MSTAR dataset are exploited, where the target, the shadow, and the clutter pixels are segmented and marked correspondingly [24]. Then, the newly constructed training and test image samples are generated by implementing the clutter transfer technique, where the clutters in the original SAR images from the MSTAR dataset are replaced with the random clutter chips from the “training pool” and the “test pool”.

The original SAR images of five different types of targets (2S1, BMP2, BTR70, T72, and ZSU23) measured at azimuth angle of 35° and elevation angles of 15° and 17° are shown in Figure 5**,** along with the clutter transfer results. It can be seen that with the clutter background modified, the appearances of the synthesized SAR images in the right column of Figure 5 are very different from the original SAR image shown in the left column of Figure 5, which would undoubtedly affect the performance of a CNN-based SAR-ATR algorithm that has only been trained to manage homogeneous clutters. Note that we are considering here the pixelwise clutter transfer, which is very different from placing multiple square-shaped target chips at random locations on a large-scene SAR image corresponding to a “presumed” observation scenario.

Once the new training and testing datasets accounting for various backgrounds are constructed, we are ready to investigate the effect of clutter background on SAR target classification. In Section 4, we demonstrate with numerical simulations that neural networks trained only with image samples with ideal backgrounds respond poorly to the background changes in the test samples. In contrast, by employing training data with various clutter backgrounds, the neural networks can learn to focus on what matters and ignore the irrelevant information.

## 3. Open-World SAR-ATR Algorithm Development and Performance Evaluation

In this section, we consider an open-world environment in which not all the test samples are from one of the known classes encountered in the network training process. We employ adversarial outlier exposure (AdvOE) proposed in [17] for network training. Consider a DNN model with parameter *θ*. Define *x* as the input image, *y* as the truth-encoded label distribution, *f*(*x*;*θ*) = softmax(*g*(*x*;*θ*)) as the normalized output vector obtained with the softmax function, and H(f(x;θ),y) as the cross-entropy loss between the predicted probabilities and *y*. The output of the DNN is a logit vector over the set of C=10−J in-distribution (ID) classes, with *J* being the number of the holdout/OOD classes. First, we construct a training dataset consisting of unlabeled OE training samples $D_{ood}^{OE}$. To minimize H(f(x;θ),y) in the 10-*J* class ID training dataset $D_{id}$ while encouraging the DNN model to output a uniform distribution over the classes $U_{C}$ for samples from $D_{ood}^{OE}$, a reasonable training objective can be formulated straightforwardly as [25]:(3)$$\underset{\theta }{\min}\underset{(x,y)\sim D_{\mathrm{id}}}{E}\left[H(f(x;\theta ),y)\right]+\lambda \underset{\tilde{x}\sim D_{\mathrm{ood}}^{\mathrm{OE}}}{E}\left[H(f(\tilde{x};\theta ),U_{C})\right],$$
which is referred to as the OE method; *λ* in Equation (3) is a user-parameter adjusting the weights put on OOD sample detection and ID sample classification. To improve the robustness of the DNN model against the potential adversarial attacks, Inkawhich et al. proposed the advOE method in [17] by incorporating adversarial training (AT) into the OE method. Specifically, a perturbation *δ* from an allowable set S_id_ is applied to the input image *x*, and the training objective in Equation (3) is modified as:(4)$$\underset{\theta }{\min}\underset{(x,y)\sim D_{\mathrm{id}}}{E}\left[\underset{{}_{\delta \in S_{\mathrm{id}}}}{\max}H(f(x+\delta ;\theta ),y)\right]+\lambda \underset{\tilde{x}\sim D_{\mathrm{ood}}^{\mathrm{OE}}}{E}\left[\underset{{}_{\delta \in S_{\mathrm{ood}}}}{\max}H(f(\tilde{x}+\delta ;\theta ),U_{C})\right].$$

To constrain the total perturbation to a maximum allowable value, it is assumed that the *l_p_* norm of *δ* is bounded by a parameter ε, i.e., ||*δ*||*_p_* ≤ ε [15].

In this work, we adopt the advOE method in Equation (4) to train the DNN model for ID sample classification. Two scenarios are considered: (1) the *standard* scenario, where the SAR images from the SAMPLE dataset are used as the ID samples just like [17]; (2) the *open-world* scenario, where the clutter transfer technique is employed to test the effect of clutter background on the OOD detection performance. Since the SAMPLE dataset only consists of 806 training samples and the clutter backgrounds of the synthesized SAR images are noticeably weaker than their measured counterparts, we augment the training data by incorporating SAR images with multiple contrast levels (see Section 4.4 for details). Additionally, we construct $D_{ood}^{OE}$ with 2491 SAR images from the SAR-ship dataset and the MiniSAR dataset, which is only 4% of the number of training samples used in [22]. Some MiniSAR image examples for nine types of military targets are shown Figure 6, which include three main battle tanks (62-HT, 63AT, T34-85), a howitzer (54TH), an anti-aircraft gun (59AG), a cannon (59-1TC), an armored personnel carrier (63APTV), an amphibious armored vehicle (63CAAV), and a fighter (J6). The resolution of each SAR image chip is 0.1 m × 0.1 m. Moreover, although the image chips from the CVDome [26], the MNIST, and the CIFAR10 datasets are used as the OOD test samples in [17], we employ the *MSTAR-O* and the *MSTAR-P* instead. Both datasets consist of 1290 SAR images corresponding to five types of targets (BRDM2, BTR60, D7, T62, ZIL131) included in the MSTAR dataset. However, the SAR image samples in MSTAR-O are self-generated with Matlab based on the official MSTAR raw data, while those in MSTAR-P are postprocessed “.jpeg” images downloaded directly from an online personal blog. The difference between the SAR images in these two datasets is illustrated in Figure 7. It is easy to observe that the clutter information of the SAR image in MSTAR-P is partially eliminated and the shadow area is obscured.

To realize OOD sample detection, two recently proposed OOD detectors, the ODIN detector and the Mahalanobis detector, are considered. The key idea of ODIN is to separate ID samples from OOD samples based on a score via temperature scaling as [17,27,28]:(5)$$S_{\mathrm{ODIN}}(x)=\underset{{}_{i\in C}}{\max}\frac{\exp(g_{i}(x)/T)}{\sum_{j=1}^{C}\exp(g_{j}(x)/T)}.$$
where $S_{\mathrm{ODIN}}(x)\in [1/|C|,1]$ is the ODIN score, and *T* is the temperature scaling parameter. In [27], *T* is set as 1000. If S_ODIN_ is greater than a predetermined threshold for a certain test sample, it reflects high-confidence prediction. In this case, the sample is considered ID, and a class label is assigned accordingly. Otherwise, the sample is declared to be OOD and rejected. With *T* set as 1, Equation (5) reduces to the *baseline detector* presented in [25], which is used as the benchmark for OOD detection performance evaluation in Section 4.4. In contrast, the main idea for the Mahalanobis detector proposed in [28] is that the OOD samples should belong to the low-density regions of the training data’s feature distributions conditioned on target class. Therefore, the OOD score for Mahalanobis detector is calculated based on the proximity of an input sample’s feature to the closest class distribution measured by Mahalanobis distance, and it is expected that the ID test samples should reveal themselves via the proximity to the training data’s feature distributions.

Finally, we introduce two performance metrics for OOD detection: the area under the receiver operating characteristic (AUROC) and the true negative rate corresponding to the threshold set to achieve a 95% true positive rate (TNR@95TPR), which are also adopted in [17]. The AUROC describes the tradeoff between true positive rate and false positive rate, which is independent of the detection threshold, while the TNR@95TPR represents the TNR when the threshold is set to achieve 95% TPR at identifying ID samples. In Section 4.4, the AUROC, TNR@95TPR, and ID sample classification accuracy are used together to evaluate the performance of the OOD detection methods and the fitness of the designated OE training sample datasets.

In [17], Inkawhich et al. demonstrated that when the SAR images from the SAMPLE dataset are used as the ID training and test data, the ODIN detector outperforms the Mahalanobis detector in detecting the holdout OOD data measured by the TNR@95TPR. In Section 4.4, however, we demonstrate with numerical simulations that when the ID training data from the SAMPLE data set are augmented with the contrast-based data augmentation method and the Mini-SAR images are used as the OE training samples, the Mahalanobis detector actually offers higher TNR@95TPR than the ODIN detector in most cases. It seems that the DNN’s capability to detect OOD samples in the feature space is enhanced by the additional ID training data from which it can learn from.

## 4. Simulation Results

This section is divided into four parts. In Section 4.1, the SAR-ATR performance improvement brought by the phase-interpolation-based SAR-target-feature synthesis technique presented in Section 2.1 is demonstrated with numerical simulations. In Section 4.2, experiments are carried out to prove that by synthesizing training data with various clutter backgrounds and the clutter transfer technique proposed in Section 2.2, the robustness of the DNNs against the background changes in the test samples improves accordingly. Note that the SAR images from the MSTAR dataset are employed for the numerical simulations in both Section 4.1 and Section 4.2. In Section 4.3, we consider a challenging scenario where the measured data from field experiments are unavailable, and only the computer-generated synthetic SAR images are used for DNN training. Exploiting the SAMPLE dataset, the boosting effects of the contrast-based data augmentation technique proposed in Section 3 on target classification accuracies are demonstrated. Finally, in Section 4.4, simulation results regarding OOD detection are provided. 

Two datasets were used in this section, the MSTAR dataset and the SAMPLE dataset. The MSTAR dataset can be downloaded from the official website (https://www.sdms.afrl.af.mil, accessed on 28 November 2022). Unfortunately, the authors do not have the permission to distribute the public SAMPLE dataset. To obtain access to the SAMPLE dataset, researchers need to contact the authors of [8]. However, for the convenience of interested readers, the image samples we presented in Section 2.2 and used to generate the simulation results in Section 4.2 have been uploaded to Github (https://github.com/gengzhe2015/SAR-target-recognition, accessed on 11 January 2023). Moreover, readers who are interested in implementing the network models presented in this section are referred to the Python codes posted by the first author of [15,17] on https://github.com/inkawhich/synthetic-to-measured-sar (accessed on 28 November 2022) and https://github.com/inkawhich/ood-sar-atr (accessed on 28 November 2022).

### 4.1. Data Augmentation with Phase Interpolation

In this subsection, we demonstrate the effectiveness of the phase-interpolation-based SAR target feature synthesis technique presented in Section 2.1. Five target types from the MSTAR dataset are considered, with the SAR images collected at elevation angles of 17° and 15° used as the training and the testing data, respectively. To simulate a scenario when the measured samples are scarce, we construct a new dataset, MSTAR-R, by choosing 136 out of 1295 measured SAR images included in the MSTAR dataset. Theoretically, one image sample could be augmented as many as 49 × 4 = 196 complex-valued images in the ±6° neighborhood of that sample’s azimuth angle, with the quality of the synthesized images deteriorating as the interpolating angle increases. To reduce the computational burden, we construct two medium-sized training datasets, MSTAR-Aug1 and MSTAR-Aug2, by choosing synthetic images within the ±1° neighborhood of the measured sample’s azimuth angle. The number of training and test samples for each type of target in MSTAR-Aug1 and MSTAR-Aug2 are summarized in Table 2.

Next, we demonstrate the performance improvement of some commonly used light-weighted neural networks provided by the data augmentation. Three networks are considered, the ResNet-18 (Res18), the AConvNet (AConv), and the SMPL [16]. Each model is trained for 60 epochs with the ADAM optimizer and an initial learning rate of 0.001, which decays at epoch 50 to 0.0001. The input images are center-cropped to 64 × 64 pixels, and the batch size is set as 128. The five-class classification accuracies provided by each model with MSTAR-R, MSTAR-Aug1, and MSTAR-Aug2 as the training dataset are presented in Table 3. Two types of loss functions are considered: the adversarial training (AT) method with ε = 2 and the label-smoothing (LSM) method with lblsm = 0.1 [15]. All results are averaged over 10 iterations. It can be seen that although the classification accuracies of AConv and SMPL are merely in the range 60–80% when the MSTAR-R is used as the training dataset, the accuracies jump to 95–98% when the augmented training dataset MSTAR-Aug1 is used. Once there are enough training samples for feature extraction, including more training data does not help in further performance improvement. Therefore, although the number of augmented data samples in MSTAR-Aug2 is approximately twice that in MSTAR-Aug1, the classification accuracies corresponding to the two training datasets are almost the same.

### 4.2. Clutter Transfer Experiment with MSTAR Dataset

In this section, we test the effect of varying clutter backgrounds on the performance of the neural networks with the MSTAR dataset. The SAR images corresponding to elevation angles of 15° and 17° are used for network training and testing, respectively. The performance degradation caused by the clutter background change in the test samples and the performance improvement achieved by incorporating training samples with diverse clutter backgrounds are summarized in Table 4. “MSTAR_OR_” represents the case in which both the training and the test samples are untouched. “Train_OR_ + Test_CT_” represents the case in which the clutter backgrounds for the test samples are randomly modified with the clutter transfer technique, while the original SAR image samples from the MSTAR dataset are used for network training. It can be seen that having been trained with samples with homogeneous clutter backgrounds, all the networks experience dramatic performance degradation when tested against samples with various clutter backgrounds (see Figure 5 for some examples). “Train_CT_ + Test_CT_” represents the case in which the clutter backgrounds of both the training and test samples are randomly chosen. It is shown that by introducing training data with various clutter backgrounds, the classification accuracies provided by all the networks bounced from the range between 50% and 60% back to higher than 90%. “Train_CT×2_ + Test_CT_” represents the case in which the clutter background of each training sample is altered in two different ways, which increases the number of training samples to twice its original quantity. It is shown that by doubling the training data samples with the clutter transfer technique, the classification accuracies provided by all the networks become approximately identical to the baseline case “MSTAR_OR_”. Through these experiments, the importance of taking into consideration the effect of the clutter background for network training and testing when addressing the SAR-ATR problem becomes obvious.

### 4.3. Data Augmentation by Employing Multiple Contrast Levels

Since there are only a small number of samples for each type of target in the SAMPLE dataset (see Table 5), we employ a novel technique for training data augmentation. Consider that the clutter background of the synthesized SAR images are noticeably weaker than their measured counterparts; we produce SAR images with three different contrast levels for network training based on the complex SAR image data matrix, which are shown in Figure 8. With data augmentation, 806 × 3 = 2418 training data samples are obtained. Next, we demonstrate the boosting effects of the proposed contrast-based data augmentation technique on target classification accuracies with numerical simulations. Define *K*
∈ [0, 1] as the fraction of images in the training set that are measured, i.e., *K* = 1 corresponds to the case in which all the training data are obtained from field experiments, while *K* = 0 refers to the case in which the DNNs are trained with 100% computer-generated synthetic SAR images and tested on measured data. Four representative light-to-medium DNN models that produced promising results for SAR-ATR are evaluated: ResNet-18, AConvNet, SMPL [16], and Heiligers’ CNN [29]. For SMPL and A-ConvNet, we set Gaus = Drop = 0.3, lblsm = 0.08, while for ResNet18 and Heiligers’ CNN, we set Gaus = Drop = 0.4, lblsm = 0.1 for optimum performance. 

The classification accuracies provided by these networks before and after training data augmentation, which are represented by “SAMPLE (Ori.)” and “SAMPLE (Aug.)”, respectively, are summarized in Table 6 for *K* = 0, 0.05, and 0.1. The improvement in classification accuracy brought by the proposed data augmentation technique is prominent, about 5% on average. It can be seen that with data augmentation implemented, the ResNet18 offers the highest classification accuracy for *K* = 0 (94.5%), *K* = 0.05 (98.1%), and *K* = 0.1 (98.9%), among all the networks under evaluation. Additionally, the LeNet-style SMPL and A-ConvNet exhibit similar performance and provide accuracies slighter lower than that of the ResNet18. Although Heiligers’ CNN produces the lowest accuracy among the DNN models for *K* ≤ 0.1, its performance is the same as other networks (approx. 100%) when *K* is large.

Note that the problem of DNN training on 100% synthetic data from the SAMPLE dataset was previously studied in [15], where Inkawhich et al. used the ensemble method with a soft-voting scheme. The predictions made by each of the five ensemble components that correspond to five different loss functions (cross-entropy, lblsm, mixup, cosine loss, AT) are averaged, and the class-labels are designated according to the highest average confidence. Although this method has been proven to be very effective in enhancing the SAR-ATR performance, it also leads to higher computational cost than using a single loss function. It is also worth mentioning that although the AT method provides higher accuracy than the other four methods, it involves adversarial attacks in multiple iterations and is strikingly more time-consuming. By using only one type of loss function (label-smoothing), our method is much more time efficient.

### 4.4. OOD Detection

In this subsection, simulation results regarding OOD detection are presented. Two scenarios are considered, the *standard* scenario that has been considered in [17], and the *open-world* scenario where the clutter transfer technique is employed to test the effect of clutter background on OOD detection performance.

#### 4.4.1. Standard Scenario

In this case, the SAR images from the SAMPLE dataset are used as the ID samples. With 2048 SAR images from the SAR-ship dataset and 443 SAR images from the MiniSAR dataset as the OE training samples (i.e., *OE training dataset #1*), the AUROC and TNR@TPR95 obtained with three different OOD detection methods, i.e., the *baseline* detector with the standard softmax threshold [25], the ODIN detector [27], and the AdvOE detector (ε = 8) based on the Mahalanobis distance for *J* = 1 and *J* = 3 are compared in Figure 9a,b, respectively. “Holdout” represents the case in which *J* classes of the targets included in the SAMPLE dataset are held out from the training dataset. The results presented for *J* = 3 are the average of 10 cases in which three types of targets are held out from the training dataset (which are the same as the ones that will be shown in Figure 11). It can be seen that the AdvOE detector based on the Mahalanobis distance offers the highest AUROC and TNR@TPR95. Moreover, it is also shown that the OOD samples from the MSTAR-P dataset are easier to detect than the MSTAR-O images, which indicates that the clutter background has a great impact on the granularity of SAR images and plays a significant role in OOD sample detection.

In the following, we illustrate how the choice regarding OE training samples affects the performance metrics for OOD sample detection. With samples from OE training dataset #1 employed, the AUROC, TNR@95TPR, and classification accuracy corresponding to different holdout class choices for J = 1 are plotted in Figure 10a. Classes #0–#9 correspond to 2S1, BMP2, BTR70, M1, M2, M35, M548, M60, T72, and ZSU23, respectively, under the assumption that K = 0.1. It can be seen from Figure 10a that when #6 (M548) and #8 (T72) are set as the holdout classes, the average TNR@95TPR are only 19% and 23%, respectively. Next, we introduce 1146 SAR images of D7 and ZIL131 as the extra OE training samples (i.e., OE training dataset #2) and plot the performance metrics in Figure 10b. It can be seen that although the TNR@95TPR for most of Holdout ID increase noticeably compared with the results shown in Figure 10a, particularly for Holdout ID #0, #3, #5, #6, and #8; the TNR@95TPR for Holdout ID #9 decreases by about 8%. It indicates that introducing more OE training samples could have either positive or negative effects on the TNR@95TPR depending on which class was held out, especially when the granularity of the additional OE samples highly resembles that of the ID samples and 90% of the ID training samples are synthetic SAR images of varying quality for different target classes. Since TNR@95TPR measures the TNR for the OOD dataset to achieve 95% TPR, the similarity between the ID and the OOD samples could lead to confusion regarding “what makes a sample ID or OOD”. Although the average TNR@95TPR obtained with OE training dataset #2 is 4% higher than that for OE training dataset #1, we apply OE training dataset #1 in the following experiments to reduce the time cost and computational burden for network training.

To further investigate this phenomenon, the performance metrics corresponding to different holdout class choices are plotted in Figure 11a,b for *J* = 2 and *J* = 3, respectively. It is easy to notice from Figure 11a that the TNR@95TPR approaches 100% when #5 (M35) and #6 (M548) are used together as the holdout classes, which is much higher than the other cases. It is an interesting result since M35 (wheeled) and M548 (tracked) are the only two trucks in the SAMPLE dataset, and the other targets are tanks (M1, M60, T72), armored personnel carrier (BMP2, BTR70, M2), artillery (2S1), and air defense (ZSU23). This trend could also be observed from Figure 11b, where the TNR@95TPR is above 90% for three cases in both figures, which correspond to holdout classes (3, 5, 6), (4, 5, 6), and (5, 6, 7), respectively. It indicates that as long as #5 (M35) and #6 (M548) are kept out of the training dataset together, they would be classified as OOD samples with high confidence. However, if only one of the two is held out, the TNR@95TPR is not necessarily high.

**Figure 11 sensors-23-00941-f011:**
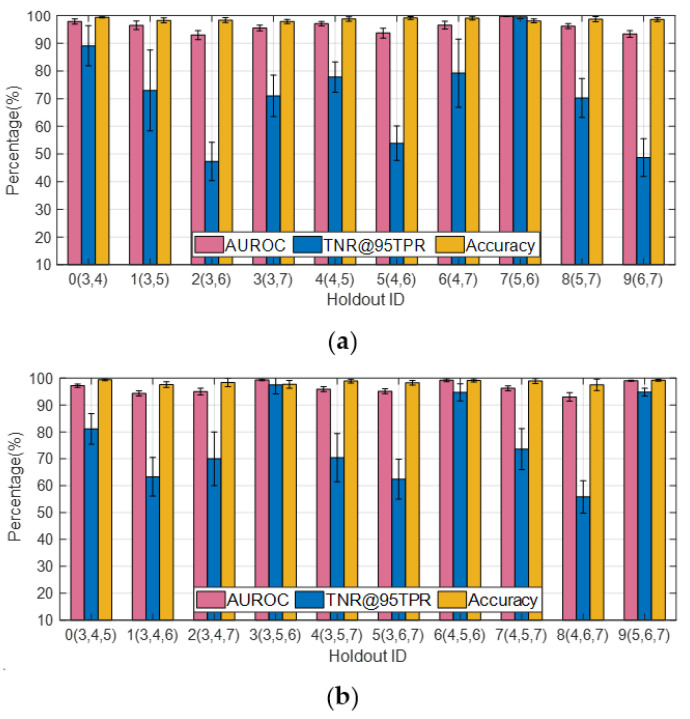
Effect of holdout class choice on detection statistics (*K* = 0.1). (**a**) *J* = 2; (**b**) *J* = 3.

Another possible reason behind the unbalanced results associated with each distinctive choice of holdout class is that some targets just have very different appearance from the other targets. For example, although both #5 (M35) and #6 (M548) are trucks, the former is much more distinguishable by appearance. The striking difference between the SAR images of M35 and M548 are illustrated in Figure 12, which also reflects the quality issue of some synthesized SAR image samples.

Finally, the effect of *K* on the detection metrics AUROC and TNR@95TPR corresponding to different holdout class choices for *J* = 3 are plotted in Figure 13a,b, respectively, with OE training dataset #1 employed. It can be seen that the AUROC improves dramatically when K increases from 0 to 0.1. Once *K* is increased to 0.5, the AUROC for all the cases surpass 98%, while the performance difference for *K* = 0.5 and 1 are trivial. When *K* = 0, the AUROC is above 85% for three cases, which correspond to holdout classes (3, 5, 6), (4, 5, 6), and (5, 6, 7), respectively.

#### 4.4.2. Open-World Scenario

To illustrate the effect of clutter on OOD detection, we consider three special cases with the holdout classes set as HLD1 = {#3: “M1”}, HLD2 = {#5: “M548”; #6: “M60”}, and HLD3 = {#3: “M1”; #5: “M548”; #6: “M60”}, respectively. Note that these cases correspond to the holdout class combinations that produce the highest AUROC, TRN@95TPR, and accuracy shown in Figure 10 and Figure 11**.** The clutter backgrounds for the SAR image samples of 2S1, BMP2, BTR70, T72, and ZSU23 taken at the elevation angle of 17° are randomly replaced by the clutter data in the MSTAR dataset, which serve as the new “measured” images for network testing. Since in this case the major difference between the synthesized and the “measured” images becomes the homogeneity level of clutter, the training data are no longer augmented by introducing multiple contrast levels, i.e., we only use the 806 training samples listed in Table 5. The performance degradation due to clutter variation for *K* = 1 are summarized in Table 7, where “SAMPLE” and “Train_OR_ + Test_CT_” represent the results for the original SAMPLE dataset and the modified dataset composed of test samples with random clutter background changes, respectively. It can be seen that the AUROC/TNR@95TPR for HLD1, HLD2, HLD3 drops dramatically from 99–100% to 83.0%/20.6%, 95.1%/67.0%, and 91.4/53.9%, respectively. “Train_CT_ + Test_CT_” and “Train_CT×2_ + Test_CT_” represent two levels of data augmentation. The former refers to the case in which 100 SAR image samples with diverse clutter backgrounds for each target type are produced for network training, while the latter refers to the case in which 200 training samples are generated for each target type by using two alternative clutter backgrounds for each image sample. Although with training data augmentation applied the resulting AUROC and TNR@95TPR remain lower than the baseline case where the backgrounds of the test samples are untouched, the performance improvement is still impressive.

## 5. Conclusions

To effectively improve the target classification accuracy offered by the DNNs in the situations where the available measured training data are scarce, novel methods are proposed in this work to augment the SAR image samples included in the MSTAR and SAMPLE datasets. SAR images corresponding to new azimuth angles are synthesized via sparse representation and phase interpolation. Hundreds of clutter chips are generated with measured clutter data included in the MSTAR dataset, and the clutter transfer technique is employed to synthesize SAR images with various clutter backgrounds. Simulation results show that when trained only with SAR image samples with the ideal homogeneous clutter backgrounds, the DNNs respond poorly to the background changes in the test samples for both ID target classification and OOD sample detection. It has also been proved that introducing training samples with diverse clutter backgrounds to the network training process leads to improved robustness and adaptivity against the varying clutter features in the test samples, even though the clutter backgrounds used to synthesize the new training and test image samples are completely different. To successfully complete the SAR-ATR missions in the open-world environment where all kinds of natural and cultural objects (e.g., trees, buildings, etc.) might be present, the DNN model needs to be capable of handling test samples with heterogeneous clutter backgrounds. Since the information content per each pixel for SAR images is much less than that for the optical imagery and corresponds to the reflectivity attributes of the target of interest and its surroundings, it is very important to train the DNN models with SAR image samples with diverse clutter data measured in field experiments rather than employing simple image perturbation or style-transfer techniques. 

To realize SAR-ATR in open environments, offline neural network training with a limited number of annotated SAR image samples for OOD sample detection is far from enough. Here, we point out two potential research directions in this field. First, the problem of super-class labeling for OOD sample needs to be solved by jointly exploiting the measured and the synthetic SAR imagery as well as multiview information fusion. According to NATO AAP-6 Glossary Terms and Definitions, “recognition” is about super-class labeling (i.e., tank), “identification” is fine-labeling (T72), while “characterization” involves specifying the subclass variants (i.e., T72-32A). It is necessary for a network trained with SAR imagery for T72 and M1 to recognize that T62 and M2 are also tanks, even if the network has never seen T62/M2 in the offline network training process. Regarding this research line, we have submitted a paper to the 2023 International Radar Symposium (IRS), which is titled as “Super-class labeling for out-of-library targets with deep learning and multiview information fusion”. Second, we could resort to active learning, cross-domain transfer learning, and transductive learning to compensate for the lack of annotated SAR training samples, and organize the scene recognition and the OOD sample detection problem into a single framework [30]. Regarding this research line, we have submitted a paper to the 2023 International Geoscience and Remote Sensing Symposium (IGARSS), which is titled as “SAR image scene classification and out-of-library target detection with cross-domain transfer learning”. We wish that by leveraging the rapid development of SAR imagery synthetization technology and the newly proposed active learning and transfer learning techniques, the bottleneck problems limiting the practical application of the DL-based SAR-ATR algorithms in real-world scenarios could be solved within this decade.

## Figures and Tables

**Figure 1 sensors-23-00941-f001:**
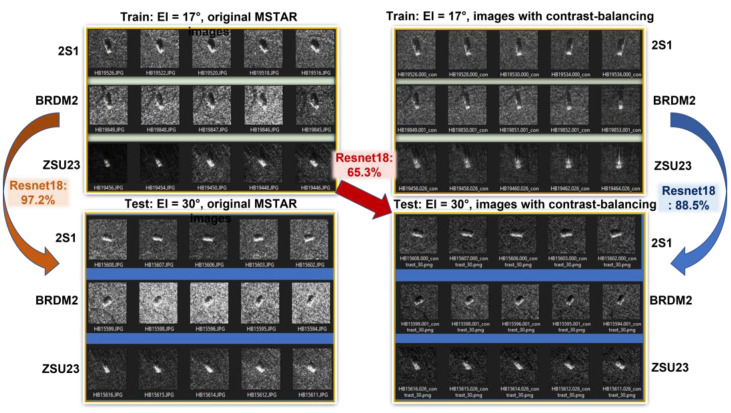
Clutter backgrounds have a great effect on target classification accuracy.

**Figure 2 sensors-23-00941-f002:**
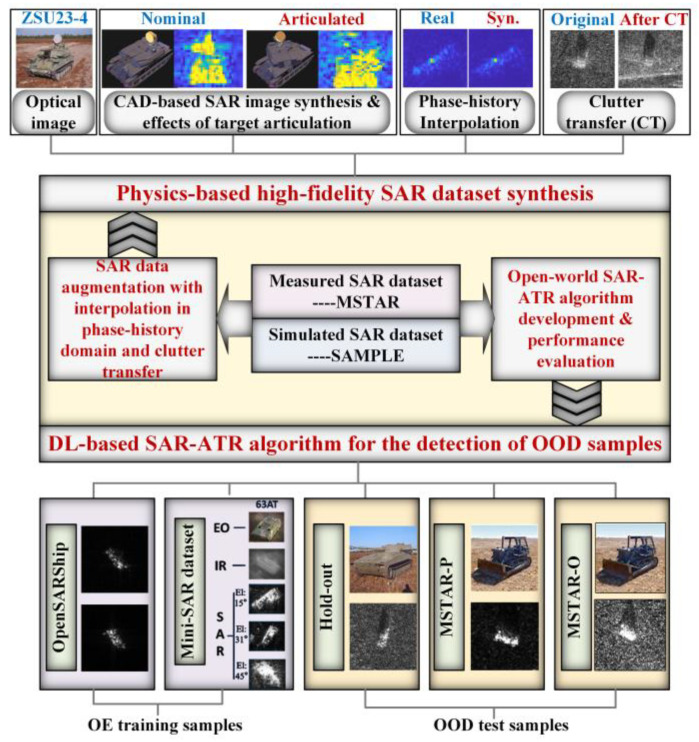
Structure of this work.

**Figure 3 sensors-23-00941-f003:**
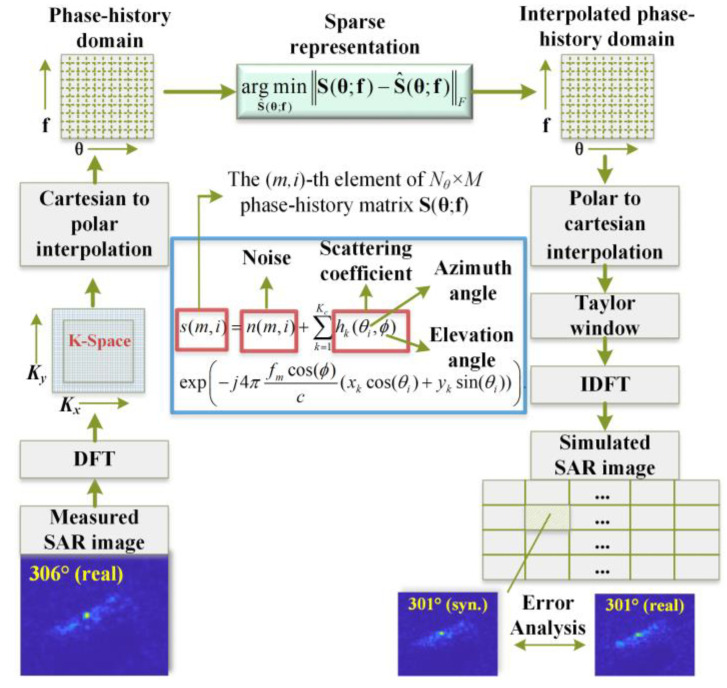
Flowchart of interpolation in the phase-history domain for SAR image data augmentation.

**Figure 4 sensors-23-00941-f004:**
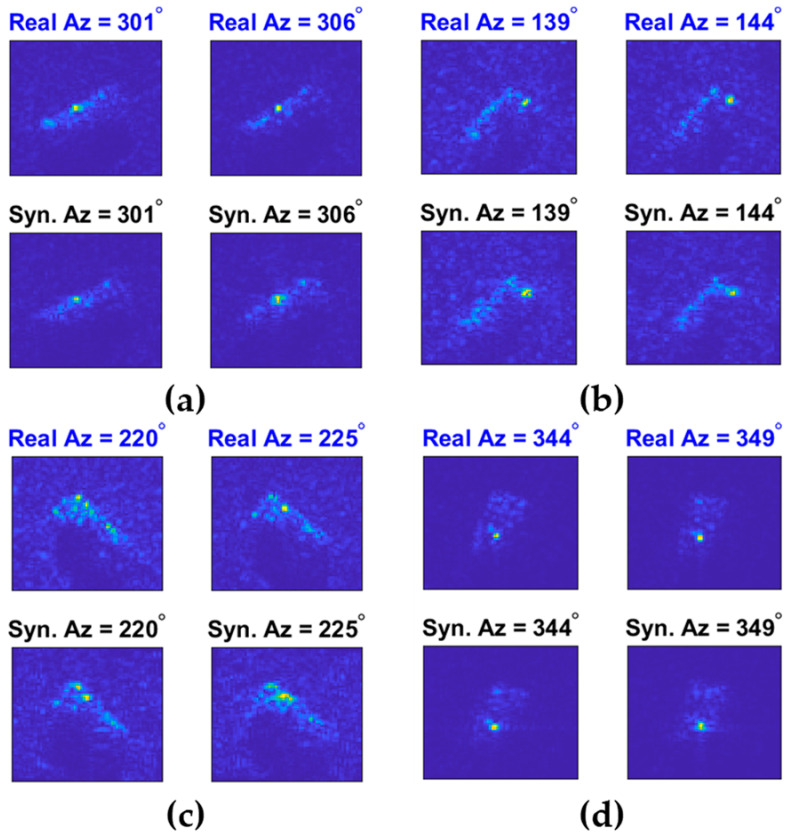
The real SAR image samples and the synthetic images obtained via interpolation (the synthetic samples at Az = θ° are obtained from the real ones corresponding to Az = θ° + 5°). (**a**) 2S1; (**b**) BRDM2; (**c**) BMP2; (**d**) ZSU23-4.

**Figure 5 sensors-23-00941-f005:**
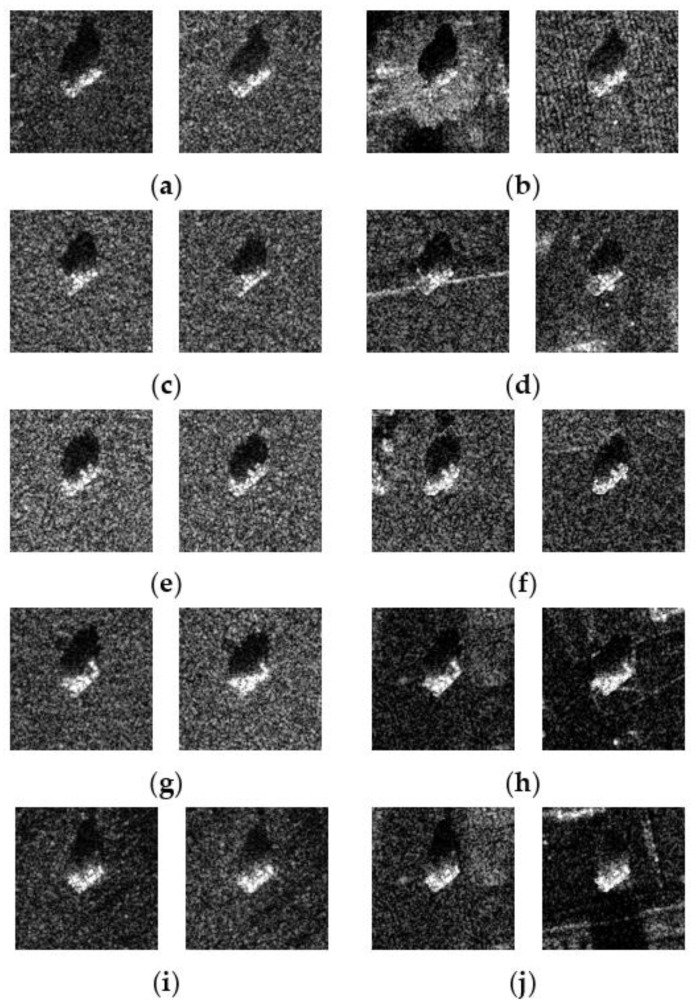
Original SAR images and corresponding clutter transfer results for five different types of targets measured at azimuth angle 48°–50° and elevation angles of 15° and 17°. It can be seen that the clutter backgrounds of the original SAR images for each type of target are highly correlated. (**a**) Original 2S1 (El =15° vs. 17°); (**b**) 2S1 with CT (El =15° vs. 17°); (**c**) Original BMP2 (El =15° vs. 17°); (**d**) BMP2 with CT (El =15° vs. 17°); (**e**) Original BTR70 (El =15° vs. 17°); (**f**) BTR70 with CT (El =15° vs. 17°); (**g**) Original T72 (El =15° vs. 17°); (**h**) T72 with CT (El =15° vs. 17°); (**i**) Original ZSU23 (El =15° vs. 17°); (**j**) ZSU23 with CT (El =15° vs. 17°).

**Figure 6 sensors-23-00941-f006:**
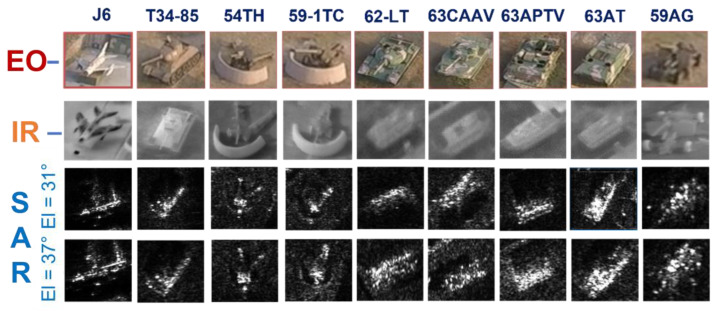
SAR images of military vehicles collected by the NUAA mini-SAR.

**Figure 7 sensors-23-00941-f007:**
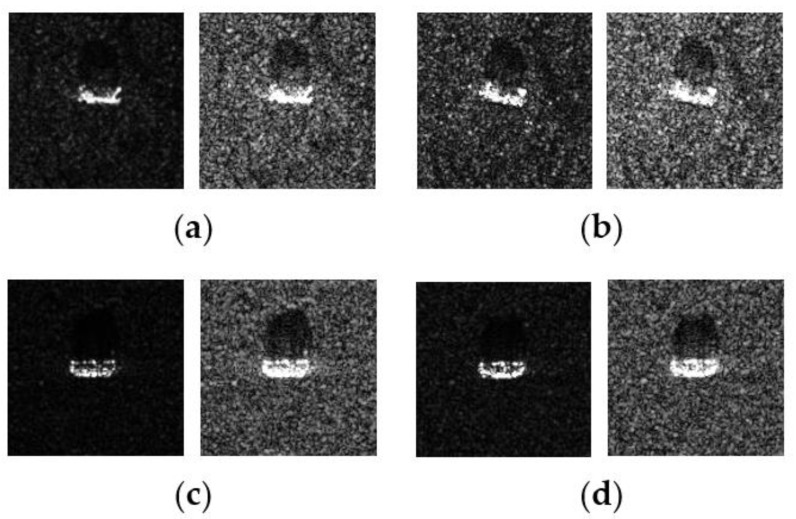
Difference between the image samples from the MSTAR-P dataset (left) and those from the MSTAR-O dataset (right). (**a**) BRDM2 (El =15°, Az = 95°); (**b**) BRDM2 (El =15°, Az = 105°); (**c**) BTR60 (El = 15°, AZ = 270°); (**d**) BTR60 (El = 17°, AZ = 270°).

**Figure 8 sensors-23-00941-f008:**
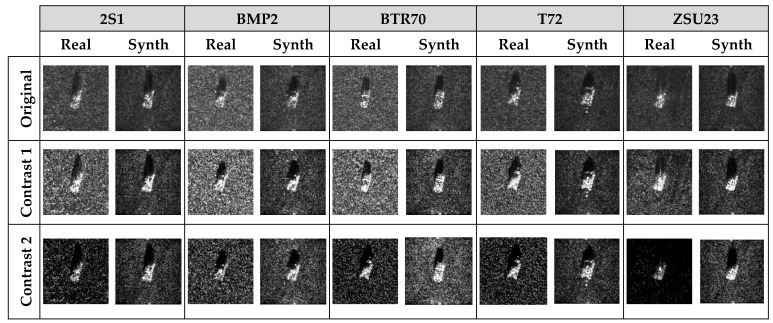
Data augmentation by introducing images with various contrast levels.

**Figure 9 sensors-23-00941-f009:**
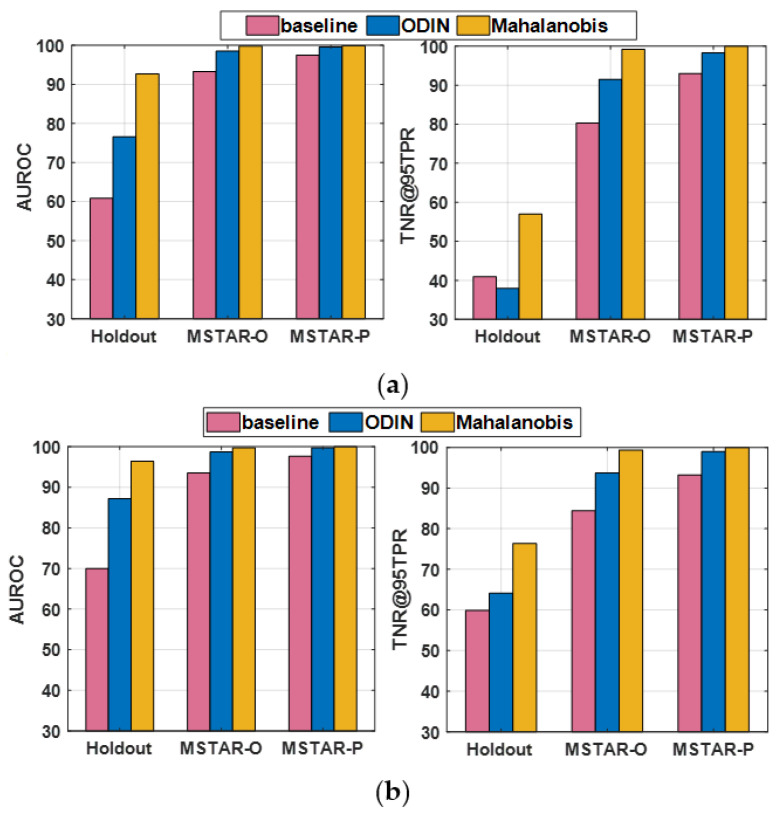
Performance comparison of different OOD detection methods. (**a**) *J* = 1; (**b**) *J* = 3.

**Figure 10 sensors-23-00941-f010:**
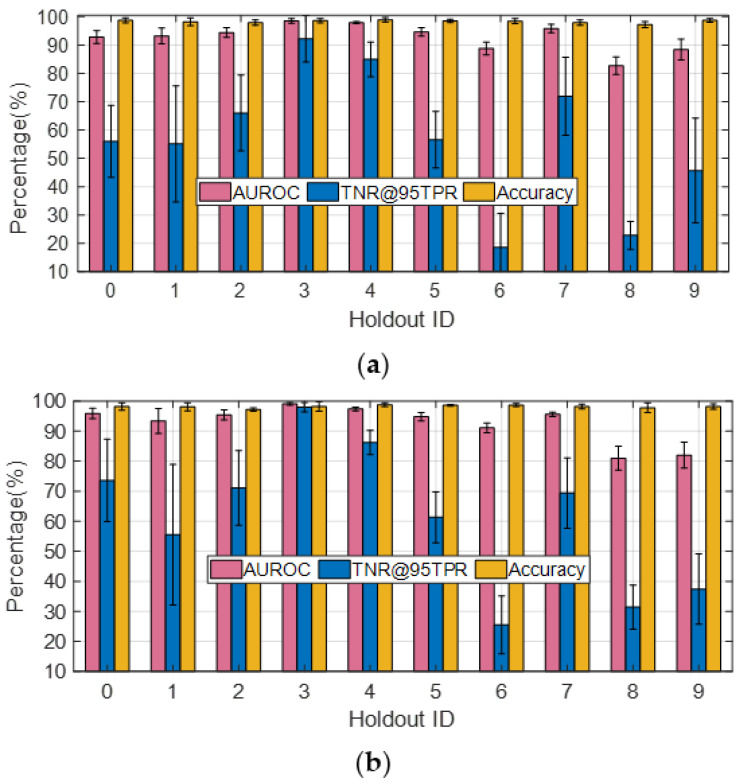
Effect of holdout class choice on detection statistics (*K* = 0.1, *J* = 1). (**a**) OE training dataset #1; (**b**) OE training dataset #2.

**Figure 12 sensors-23-00941-f012:**
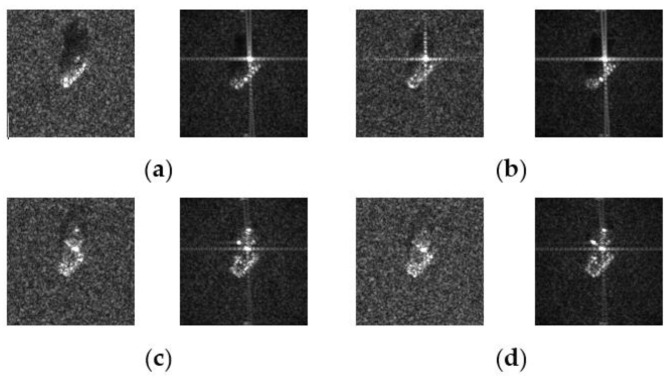
Real and synthesized SAR image samples for M35 and M548. (**a**) M35 El = 16°, Az = 37°; (**b**) M35 El = 17°, Az = 37°; (**c**) M548 El = 16°, Az =56°; (**d**) M548 El = 16°, Az =56°.

**Figure 13 sensors-23-00941-f013:**
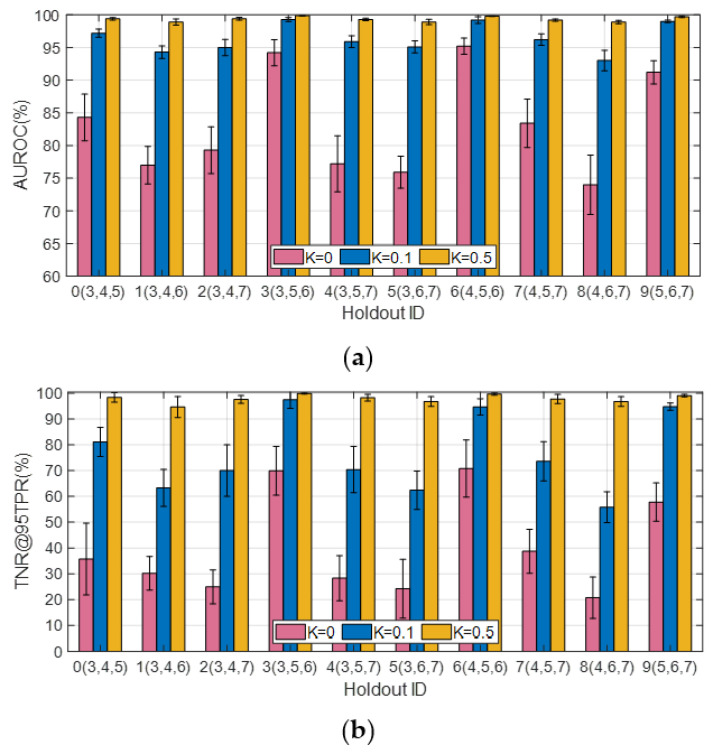
Effect of *K* on AUROC and TNR@95TPR (*J* = 3). (**a**) AUROC; (**b**) TNR@95TPR.

**Table 1 sensors-23-00941-t001:** Comparison of measured and synthetic SAR imagery in SAMPLE dataset.

Target	Measured/Synthetic	Target	Measured/Synthetic
2S1 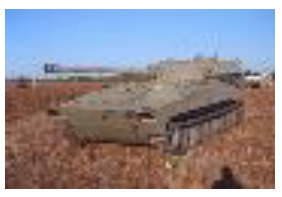	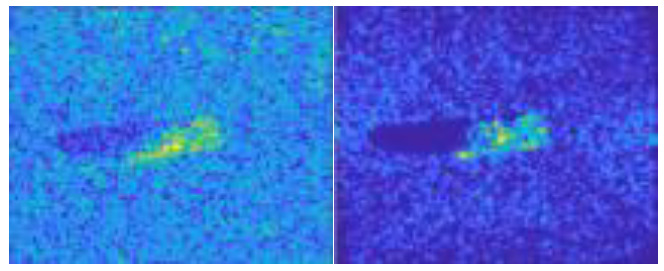	M35 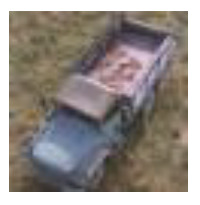	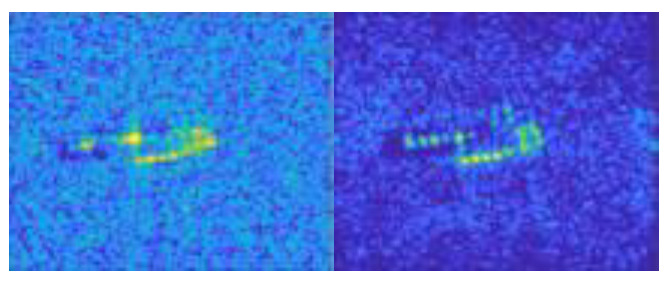
BMP2 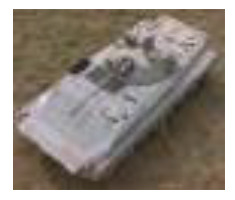	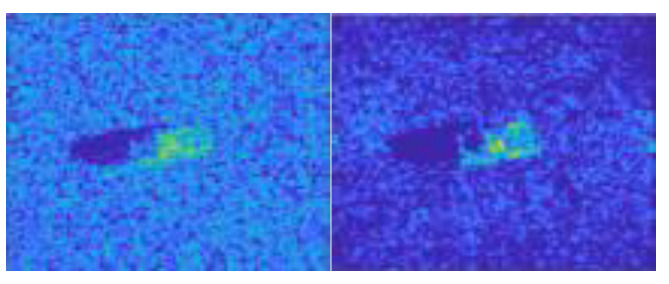	M548 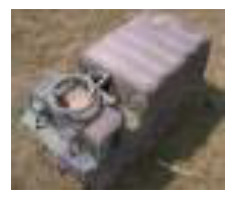	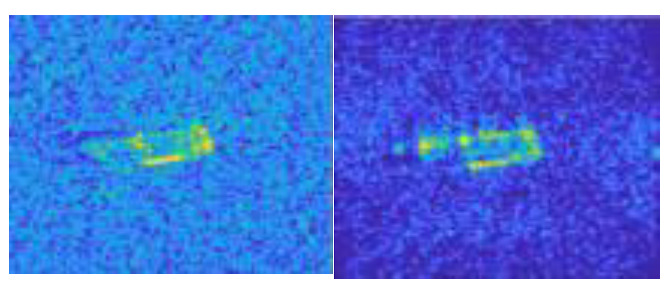
BTR70 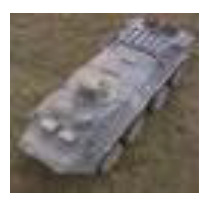	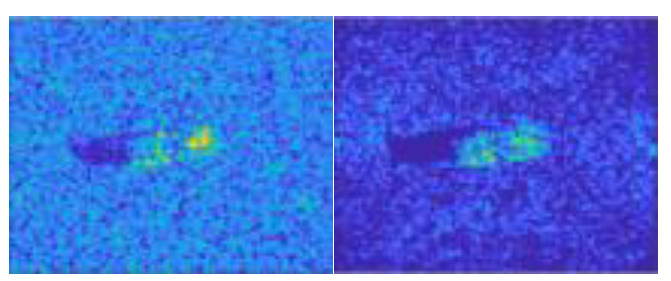	M60 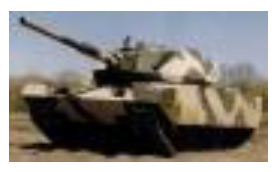	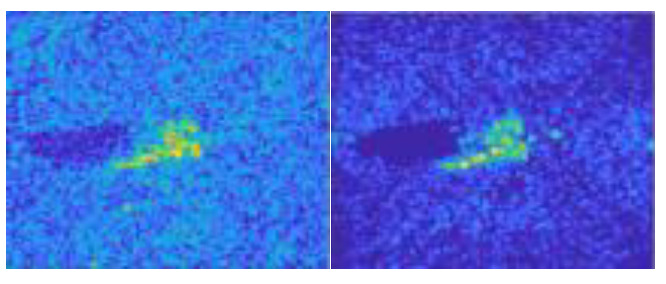
M1 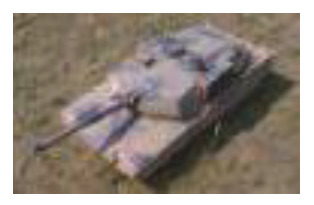	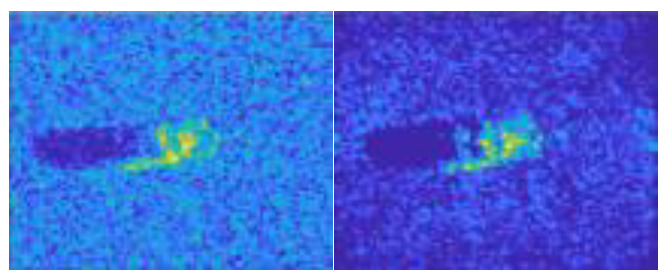	T72 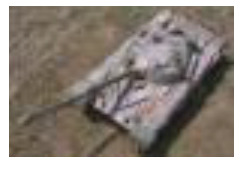	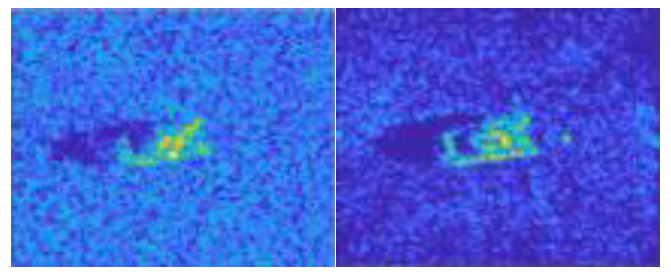
M2 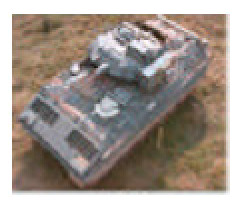	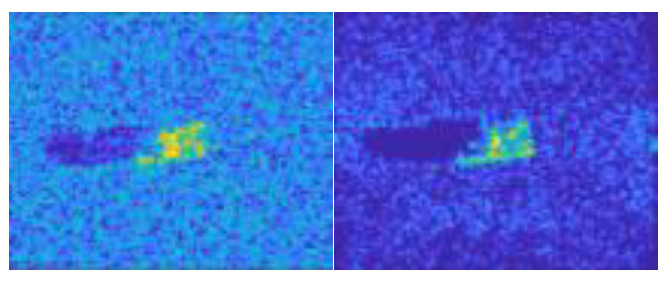	ZSU23 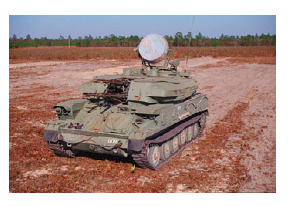	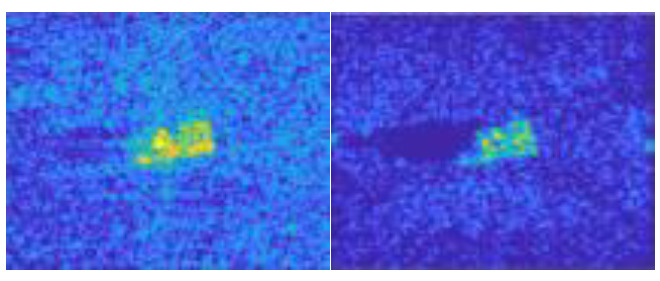

**Table 2 sensors-23-00941-t002:** Number of training image samples obtained via phase interpolation.

	MSTAR-R		MSTAR_Aug1		MSTAR_Aug2	
Class	Train	Test	Train	Test	Train	Test
2S1	32	274	256	274	448	274
BMP2	24	195	192	195	336	195
BTR70	24	196	192	196	336	196
T72	24	195	192	195	336	195
ZSU23	32	274	256	274	448	274
Number of samples	136	1134	1088	1134	1904	1134

**Table 3 sensors-23-00941-t003:** Performance improvement of the neural networks provided by the synthetic images obtained via interpolation.

Model	5-Class Classification Accuracy			Method
	MSTAR-R	MSTAR-Aug1	MSTAR-Aug2	
Res18	83.7 ± 4.35	97.4 ± 1.00	97.8 ± 0.16	AT (ε = 2)
AConv	68.3 ± 1.22	95.6 ± 0.95	96.4 ± 0.69	
SMPL	56.6 ± 2.33	96.4 ± 0.59	96.6 ± 0.64	
Res18	75.1 ± 8.80	95.1 ± 3.41	95.2 ± 2.61	LSM (lblsm = 0.1)
AConv	73.0 ± 5.37	97.0 ± 0.74	97.2 ± 0.35	
SMPL	61.3 ± 3.37	97.6 ± 0.63	97.6 ± 0.29	

**Table 4 sensors-23-00941-t004:** Improved robustness against the background changes in test samples by introducing training samples with diverse clutter backgrounds.

Model	5-Class Classification Accuracy (%)			
	MSTAR_OR_	Train_OR_ + Test_CT_	Train_CT_ + Test_CT_	Train_CT×2_ + Test_CT_
SMPL7	98.1 ± 0.72	38.6 ± 1.17	91.5 ± 0.93	96.0 ± 1.03
RN18	99.8 ± 0.06	55.2 ± 1.42	97.5 ± 0.65	98.4 ± 0.35
AConv	99.0 ± 0.21	54.2 ± 5.87	94.1 ± 0.60	98.0 ± 0.39
Heiligers	98.4 ± 0.39	54.7 ± 3.39	87.1 ± 1.58	92.5 ± 1.41

**Table 5 sensors-23-00941-t005:** Number of samples for each type of target in the SAMPLE dataset.

Class	Train	Test
2S1	116	58
BMP2	55	52
BTR70	43	49
M1	78	51
M2	75	53
M35	76	53
M548	75	53
M60	116	60
T72	56	52
ZSU23	116	58
Number of samples	806	539

**Table 6 sensors-23-00941-t006:** Effect of contrast-based data augmentation on target classification accuracies.

Model	SAMPLE (Ori.)			SAMPLE (Aug.)		
	Min	Max	Avg ± std	Min	Max	Avg ± std
*K* = 0						
SMPL7	76.4	93.5	86.7 ± 3.41	89.2	96.3	91.8 ± 2.00
RN18	85.0	95.5	91.9 ± 2.17	92.2	96.3	94.5 ± 1.32
AConv	84.4	89.1	86.5 ± 1.77	87.0	92.6	89.6 ± 1.62
Heiligers	76.6	84.6	80.4 ± 1.96	80.1	87.6	84.5 ± 2.17
*K* = 0.05						
SMPL7	82.2	92.4	89.1 ± 2.56	92.6	97.8	96.2 ± 1.44
RN18	95.4	98.5	96.9 ± 1.02	95.7	99.3	98.1 ± 1.00
AConv	86.5	93.3	90.5 ± 2.13	93.7	97.2	95.7 ± 1.05
Heiligers	83.7	90.4	86.9 ± 1.67	84.6	95.2	90.2 ± 3.24
*K* = 0.1						
SMPL7	88.1	94.1	92.0 ± 1.52	96.5	99.3	97.9 ± 0.08
RN18	96.7	98.9	97.8 ± 0.81	98.1	99.6	98.9 ± 0.54
AConv	88.7	94.6	92.5 ± 1.67	96.8	98.9	98.0 ± 0.74
Heiligers	84.8	93.3	88.4 ± 2.18	89.4	96.7	93.8 ± 1.97

**Table 7 sensors-23-00941-t007:** OOD detection performance degradation due to background changes in ID test samples, and the positive effect brought by the extra training samples with diverse backgrounds (*K* = 1).

	OOD Dataset	OOD Detection Performance (AUROC, %)		
		Holdout		
ID Dataset		HLD1	HLD2	HLD3
SAMPLE		99.6 ± 0.19	100.0 ± 0.00	99.8 ± 0.04
Train_OR_ + Test_CT_		83.0 ± 5.88	95.1 ± 1.95	91.4 ± 1.26
Train_CT_ + Test_CT_		95.2 ± 1.12	96.3 ± 0.80	93.2 ± 1.06
Train_CT×2_ + Test_CT_		97.7 ± 0.90	98.4 ± 0.64	98.4 ± 0.56
	**OOD Dataset**	**OOD Detection Performance (TNR@95TPR, %)**		
		**Holdout**		
**ID Dataset**		**HLD1**	**HLD2**	**HLD3**
SAMPLE		98.7 ± 0.92	100.0 ± 0.00	99.6 ± 0.47
Train_OR_ + Test_CT_		20.6 ± 11.4	67.0 ± 15.5	53.9 ± 6.52
Train_CT_ + Test_CT_		64.4 ± 6.64	70.3 ± 10.9	69.6 ± 5.18
Train_CT×2_ + Test_CT_		86.6 ± 5.24	89.2 ± 6.09	88.5 ± 6.09

## Data Availability

Publicly available datasets were analyzed in this study. This data can be found here: https://www.sdms.afrl.af.mil (accessed on 28 November 2022).

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
