# Peer review of "Target Recognition in SAR Images by Deep Learning with Training Data Augmentation"

_sensors, 2023, doi:10.3390/s23020941_

Round 1

Reviewer 1 Report

The proposed paper deals with a data augmentation strategy in deep learning in the SAR domain for military target recognition. The authors show clearly the interest in data augmentation. Two approaches are used. The first one consists in phase interpolation in the phase - history domain. In the second one a clutter transfer process is used; in this case, the authors show clearly that the introduction of augmentation has to be made for the training data. A modification of the clutter only in the test data is responsible of a significant drop of the efficiency. Different CNN architectures have been tested, the all show the interest in data augmentation 

Concerning the detection, two ways are investigated and an interesting discussion has been made.The OOD detection seems to be appropriate for the problem. 

Finally the strong effect of the background has been pointed out. 

This paper show very interesting result. I have no particular modifications to demand, as a result, I think this paper can be published in its present form.

Author Response

Response to Reviewer 1 Comments:

The authors appreciate the insightful comments made by Reviewer 1. We also like to thank the time and efforts Reviewer 1 put into this manuscript.

Reviewer 2 Report

Authors exploit both the widely-used Moving and  Stationary Target Acquisition and Recognition (MSTAR) SAR dataset and the Synthetic and Measured Paired Labeled Experiment (SAMPLE) dataset, which consists of selected samples from the 

MSTAR dataset and their computer-generated synthetic counterparts. A series of data augmentation experiments are carried out.

 First, the sparsity of the scattering centers of the targets is exploited for new target pose synthesis. Meanwhile, training data with various clutter backgrounds are synthesized via clutter transfer, so that the neural networks are better prepared to cope with background changes in the test samples.

The purpose of this work is to lay the foundation for the large-scale  open-field implementation of DL-based SAR ATR systems, which is not only of great value in the sense of theoretical research but also potentially meaningful in the aspect of military application.

In this paper, The major contributions are summarized as following: 

• The effectiveness of the phase-interpolation based training data augmentation technique is demonstrated with the MSTAR dataset, and a novel contrast-based method is proposed to augment the synthetic training samples in the SAMPLE dataset. 

• The impact of the clutter background on SAR target classification is evaluated by exploiting the clutter data in the MSTAR dataset. It is shown that by introducing training data with various clutter backgrounds, the algorithm is robust against the  background changes of the test samples. 

• By using the MiniSAR images as the OE training samples, diverse CNN models are designed and trained to detect OOD samples from the MSTAR-O and the MSTAR-P dataset. It is shown that the contrast-based data augmentation method is also effective in improving the OOD sample detection performance.

1. Please ensure that all variables/symbols introduced in the manuscript are properly explained and the index of each  symbol is correct and consistent in order to avoid confusion.

      2. Authors should add the open sourced code of proposed model and datasets. Information about open sourced code and github repo could help reproduce and further understand this study on a programming level would be beneficial for future research and experiments.

3. Several references about  deep learning usage and model can be added [1][2][3][4].

[1] Chao Li, Jinfan Liu, Chao Wang, Influence of Noise Level on MSTAR Images Recognition Performance, Procedia Computer Science, Volume 187, 2021, Pages 97-102, https://doi.org/10.1016/j.procs.2021.04.039.

[2] Necip Cinar, Alper Ozcan, Mehmet Kaya, A hybrid DenseNet121-UNet model for brain tumor segmentation from MR Images, Biomedical Signal Processing and Control, Volume 76, 2022, ISSN 1746-8094, https://doi.org/10.1016/j.bspc.2022.103647.

[3] Rui Lu, Xiaobo Liu, Xiaoming Chen, Localization and tracking of multiple fast moving targets in bistatic MIMO radar, Signal Processing, 2023, https://doi.org/10.1016/j.sigpro.2022.108780.

[4] A. Shakin Banu, P. Vasuki, S. Md Mansoor Roomi, Target detection in SAR images using Bayesian Saliency and Morphological attribute profiles, Computer Communications, Volume 160,2020, Pages 738-748,  https://doi.org/10.1016/j.comcom.2020.03.018.

4. The experimental analysis section should be strengthened. The authors can evaluate different architectures 

Author Response

Response to Reviewer 2 Comments

Comment 1: Please ensure that all variables/symbols introduced in the manuscript are properly explained and the index of each symbol is correct and consistent in order to avoid confusion.

Response: Thank you for the suggestion. We have carefully checked all the variables/symbols used in the manuscript.

Comment 2: Authors should add the open sourced code of proposed model and datasets. Information about open sourced code and github repo could help reproduce and further understand this study on a programming level would be beneficial for future research and experiments.

Response: Thank you for the suggestion. Two datasets were used in this work, the MSTAR dataset and the SAMPLE dataset. The MSTAR dataset could be downloaded from the official website (https://www.sdms.afrl.af.mil). Unfortunately, the authors don’t have the permission to distribute the public SAMPLE dataset. To get access to the SAMPLE dataset, the researchers need to contact the authors of [8] via email (the authors did just that and got a response from them within two days). However, for the convenience of other researchers who are interested in this work, the image samples we presented in Section 2.2 and used to generate the simulation results in Section 4.2 have been uploaded to Github (https://github.com/gengzhe2015/SAR-target-recognition).

As for implementing the neural network models with Python, the authors leveraged the codes posted by Dr. Inkawhich (i.e. the author of [15] and [17]) on https://github.com/inkawhich/synthetic-to-measured-sar and https://github.com/inkawhich/ood-sar-atr. The other researchers could regenerate the simulation results presented in this work based on these codes. We also provided a short description in the revised work regarding this point (see the 2nd paragraph of Section 4.2 of the revised paper).

Comment 3: Several references about deep learning usage and model can be added [1][2][3][4].

[1] Chao Li, Jinfan Liu, Chao Wang, Influence of Noise Level on MSTAR Images Recognition Performance, Procedia Computer Science, Volume 187, 2021, Pages 97-102, https://doi.org/10.1016/j.procs.2021.04.039.

[2] Necip Cinar, Alper Ozcan, Mehmet Kaya, A hybrid DenseNet121-UNet model for brain tumor segmentation from MR Images, Biomedical Signal Processing and Control, Volume 76, 2022, ISSN 1746-8094, https://doi.org/10.1016/j.bspc.2022.103647.

[3] Rui Lu, Xiaobo Liu, Xiaoming Chen, Localization and tracking of multiple fast moving targets in bistatic MIMO radar, Signal Processing, 2023, https://doi.org/10.1016/j.sigpro.2022.108780.

[4] A. Shakin Banu, P. Vasuki, S. Md Mansoor Roomi, Target detection in SAR images using Bayesian Saliency and Morphological attribute profiles, Computer Communications, Volume 160,2020, Pages 738-748,  https://doi.org/10.1016/j.comcom.2020.03.018.

Response: Thank you for the suggestion. However, after carefully checking the references recommended by the reviewer, we decide that these works are not very closely related to the problems addressed in this work and opt not to add them. The logic is as following. First, we composed this work without reading those reference works. Second, after reading those reference works, we didn’t find the necessity/motivation to make significant changes (i.e. research methodology, experiment results analysis, conclusion) to the original manuscript.

Comment 4: The experimental analysis section should be strengthened. The authors can evaluate different architectures.

Response: Thank you for the suggestion. We considered three architectures in Section 4.1 and four architectures in Section 4.2-4.3, which is consistent with [15]. Since it has been proved in [15] that WideResNet18 doesn’t provide superior SAR ATR performance than ResNet18, we used AConvNet as a substitute for WideResNet18. The networks considered in this work have been trained from scratches with a PC with Intel Xeon Silver 4210 CPU (64G memory) and NVIDIA GeForce RTX3070 GPU (8G memory). Due to the limitation of GPU memory, the batch size has to be set as small as 16 when training large networks (e.g. DenseNet121). However, the results for other networks are all obtained by setting the batch size as 128. Since a decision of “minor revision” was granted by the editor, we have to upload the revised work within 5 days. Due to the impact of Covid-19, the authors are all currently working remotely. It is just not practical for the authors to redo everything. We have tried our best to provide insightful experiment results analysis in the original, and we feel regretful that the reviewer found that the experimental analysis section should be strengthened.

Reviewer 3 Report

1. Abbreviations have to be provided in full name the first time they are used with the short text in brackets. Please check and apply across the document.
2. Author should compare with more recent literature and mention the specific gap in the research. 3. In the Conclusion section, authors can add 1-2 good future directions. 4. Author should check that all cited papers should be in proper order.

Author Response

Response to Reviewer 3 Comments

Comment 1: Abbreviations have to be provided in full name the first time they are used with the short text in brackets. Please check and apply across the document.

Response: Thank you for the suggestion. We have provided the full name for all the abbreviations used in this work.

Comment 2: Author should compare with more recent literature and mention the specific gap in the research.

Response: Thank you for the suggestion. We submitted our paper for review in October, 2022 and our major references are [15] and [17], which were both published in 2021. We summarized our contributions compared to [15] and [17] at the end of Section 1, which are well-supported by the simulation results we provided in Section 4.

Comment 3: In the Conclusion section, authors can add 1-2 good future directions.

Response: Thank you for the suggestion. We have added a paragraph along with a new reference work ([30]) to the Conclusion section to point out potential future research directions. Hope the reviewer found our revision acceptable.

  1. Inkawhich, N.; Zhang, J.; Davis, E. K.; Luley, R.; Chen, Y. Improving out-of-distribution detection by learning from the deployment environment. IEEE Journal of Selected Topics in Applied Earth Observations and Remote Sensing, 2022, 15, 2070–2086

Comment 4: Author should check that all cited papers should be in proper order.

Response: Thank you for the suggestion. We have checked the citation and are certain that they’re in proper order.